# Impact of Vitamin D in Prophylaxis and Treatment in Tuberculosis Patients

**DOI:** 10.3390/ijms23073860

**Published:** 2022-03-31

**Authors:** Roberta Papagni, Carmen Pellegrino, Francesco Di Gennaro, Giulia Patti, Aurelia Ricciardi, Roberta Novara, Sergio Cotugno, Maria Musso, Giacomo Guido, Luigi Ronga, Stefania Stolfa, Davide Fiore Bavaro, Federica Romanelli, Valentina Totaro, Rossana Lattanzio, Giuseppina De Iaco, Fabrizio Palmieri, Annalisa Saracino, Gina Gualano

**Affiliations:** 1Clinic of Infectious Diseases, Department of Biomedical Sciences and Human Oncology, University of Bari “Aldo Moro”, 70123 Bari, Italy; robertapapagni0@gmail.com (R.P.); karmenpellegrino@gmail.com (C.P.); giuliapatti22@gmail.com (G.P.); aurelia.r.92@gmail.com (A.R.); robertanovara@gmail.com (R.N.); sergio.cotugno@gmail.com (S.C.); giacguido@gmail.com (G.G.); davidebavaro@gmail.com (D.F.B.); valentinatotaro11@gmail.com (V.T.); rossana.lattanzio@gmail.com (R.L.); giusideiaco@gmail.com (G.D.I.); annalisa.saracino@uniba.it (A.S.); 2National Institute for Infectious Diseases “L. Spallanzani” IRCCS, 00161 Rome, Italy; maria.musso@inmi.it (M.M.); fabrizio.palmieri@inmi.it (F.P.); gina.gualano@inmi.it (G.G.); 3Microbiology and Virology Unit, University of Bari, University Hospital Policlinico, 70124 Bari, Italy; rongalu@yahoo.it (L.R.); stolfastefania@gmail.com (S.S.); federicarosarosaromanelli@gmail.com (F.R.)

**Keywords:** vitamin D, tuberculosis, TB, outcome, prophylaxis, treatment

## Abstract

Vitamin D plays a crucial role in many infectious diseases, such as tuberculosis (TB), that remains one of the world’s top infectious killers with 1.5 million deaths from TB in 2021. Vitamin D suppresses the replication of *Mycobacterium tuberculosis* in vitro and showed a promising role in TB management as a result of its connection with oxidative balance. Our review encourages the possible in vivo benefit of a joint administration with other vitamins, such as vitamin A, which share a known antimycobacterial action with vitamin D. However, considering the low incidence of side effects even at high dosages and its low cost, it would be advisable to assess vitamin D level both in patients with active TB and high-risk groups and administer it, at least to reach sufficiency levels.

## 1. Introduction

Vitamin D, also called cholecalciferol, is a fat-soluble vitamin, which plays a fundamental role in bone constitution and remodeling, influencing the reabsorption of calcium from the bone and intestine [1]. Due to its structure, it acts more like a hormone. In fact, its metabolites also have different activities, which can influence other processes such as cellular differentiation and proliferation, immune system regulation, central nervous system function and cardiovascular health [2,3]. For these functions, it seems that vitamin D plays an important role in preventing the development of tumors [4,5], cardiovascular diseases, multiple sclerosis, type 1 diabetes, infectious diseases and other conditions [6]. In particular, as regards the immune system, an important role of vitamin D metabolites has been found in immunomodulation, as they act by activating monocytes, influencing the synthesis of cytokines and immunoglobulins and suppressing lymphocyte proliferation. The evidence of this interaction is given by the presence of Vitamin D Receptor (VDR) on the membrane of circulating monocytes and activated T cells [7,8]. This relationship is not new; in fact, malnourished children are known to be more vulnerable to respiratory tract infections [9]. Other studies have proposed a relationship between sun exposure and the onset of seasonal winter diseases like influenza [10]. Vitamin D plays a crucial role in many infectious diseases, such as tuberculosis (TB), which remains one of the world’s top infectious killers with 1.5 million deaths from TB in 2021 [11].

The aim of this review is to outline the available findings as well as the unanswered issues about the role of vitamin D in the context of *Mycobacterium*
*tuberculosis* (MT) infection.

## 2. Metabolism of Vitamin D

In the skin, ultraviolet radiation causes the photochemical conversion of 7-dehydrocholesterol to vitamin D. The production of vitamin D is reduced by melanin and sun creams with a high protection factor, which effectively block the skin penetration of UV rays. In food, vitamin D is found mainly in fish oils and egg yolk. Vitamin D of vegetable origin is represented by vitamin D_2_, while that of animal origin is represented by D_3_. These two forms have equivalent biological activity and are activated by vitamin D hydroxylases in a similar way. Vitamin D, synthesized in the skin or absorbed in the intestine, circulates linked to an alpha-globulin synthesized by the liver, called vitamin D binding protein (VDBP). In the liver, vitamin D is hydroxylated to 25-hydroxyvitamin D [25(OH)D] by an oxidase-cytochrome P450, whose activity is not subject to strict regulation. 25(OH)D is the main circulating and storage form of vitamin D. About 88% of it is bound to VDBP, about 12% to albumin and a minimum share circulates freely. The half-life of 25(OH)D is approximately 2–3 weeks. In particular, the half-life of 25(OH)D_2_ is lower than that of 25(OH)D_3_, due to the lower affinity of the former for VDBP. The half-life of vitamin D is reduced when there are low levels of VDBP, such as in the case of increased loss in nephrotic syndrome [12]. In the kidney, 25(OH)D is further hydroxylated by an oxidase and then transformed into the active hormone: 1,25 dihydroxy vitamin D [1,25(OH)_2_D] or calcitriol. This oxidase is expressed in the cells of the proximal convoluted tubule and subjected to strict regulation: it is stimulated by parathyroid hormone and hypophosphatemia and inhibited by calcium, FGF23 and 1,25(OH)_2_D itself. The active hormone acts on intracellular VDR and influences gene transcription [12,13]. Figure 1 shows metabolism of vitamin D.

## 3. Vitamin D Deficiency

The diagnosis of vitamin D deficiency is based on determining the plasma concentration of 25(OH)D, as it represents the most reliable indicator of vitamin D reserves of the body, as shown in Table 1. In 2011, the USA Endocrine Society together with other Scientific Societies established the thresholds for vitamin D levels as indicated below [14].

These thresholds are related to the role of vitamin D in the regulation of mineral homeostasis and bone health. Treatment or prevention of different conditions probably requires different concentrations of the vitamin, but for the other conditions (i.e., prevention of cancer, autoimmune/cardiovascular/infection diseases, etc.), no cut-offs supported by high levels of evidence are available. However, from studies on optimal serum concentrations of 25(OH)D in relation to different conditions (bone mineral density, lower-extremity function, dental health, risk of falls, fractures and colorectal cancer), it seems that for most of these conditions, benefits are obtained starting from 30 ng/mL [15,16].

## 4. Vitamin D and Tuberculosis

Before the advent of anti-tuberculosis therapy, the benefit of administering cod liver oil and sun exposure for tuberculosis patients had been noted [17]. Moreover, N. R. Finsen demonstrated the effectiveness of short-wave UV light in the treatment of cutaneous TB, a discovery for which he won the Nobel Prize in 1903 [18]. Vitamin D appears to play an important role against MT infection through different mechanisms.

### 4.1. Innate Immunity 

MT uses the toll-like receptors (TLRs) present on the surface of macrophages to enter the body. In macrophages, following the activation of the signaling pathway mediated by TLRs and exposure to inflammatory cytokines, there is an increase in the expression of CYP27B1 oxidase, responsible for the oxidation of 25(OH)D to the active form 1,25(OH)_2_D. This, through an autocrine mechanism, activates the signaling pathway mediated by the VDR/RXR receptors present on the macrophages themselves, with a consequent increase in the synthesis of cathelicidin hCAP-18 from which the Leucine-Leucine-37 peptide (LL-37) is derived. It destroys the bacterial cell by interacting with the molecules of the bacterial wall and by perforating the cytoplasmic membrane [6,19,20,21].

Other studies have also pointed out that vitamin D seems to induce autophagy in infected macrophages [22]. In fact, if on the one hand MT tries to block the autophagic process, through which the infected macrophages eliminate the bacteria contained in the phagolysosomes, on the other hand, the mycobacterial lipoproteins activate the signaling pathways TLR2/1, CD14 and VDR, which stimulate autophagy [23]. Moreover, several studies have demonstrated the role of LL-37 in autophagy; in particular, it has been highlighted that the autophagic process is blocked if the expression of LL-37 is silenced [24]. Another study showed that the administration of vitamin D and phenylbutyrate is associated with a marked increase in LL-37 levels in macrophages and lymphocytes and with increased intracellular killing of MT [25].

Vitamin D also appears to inhibit the growth of MT in infected macrophages through the production of nitrogen and oxygen reactants: in fact, a study has shown that macrophages, under the stimulation of LPS and 1,25(OH)_2_D_3_, increase the expression of inducible nitric oxide synthase (NOS2), acquiring the ability to produce a large amount of NO [26].

### 4.2. Acquired Immunity 

Some studies have reported the absence of some antimycobacterial activities mediated by T-helper lymphocytes in case of vitamin D deficiency: for example, the production of INF-γ by Th1 lymphocytes, which enhances antibacterial activity of macrophages and stimulation of the antibody-mediated response by Th2 lymphocytes via the secretion of IL4 and IL5 [13,27].

### 4.3. Anti-Inflammatory Activity 

If on the one hand vitamin D seems to carry out all these pro-inflammatory and antimicrobial activities aimed at counteracting the MT infection, on the other, it also seems to perform anti-inflammatory functions that limit an excessive inflammatory response.

In fact, several studies have shown that vitamin D has an anti-inflammatory activity, through various mechanisms, including the induction of the expansion of T-reg lymphocytes, which in turn limit the activity of Th1 and the regulation of the expression of the genes that encode for metalloproteinases (MMPs) [28]. MMPs are known to be associated with tissue remodeling and the formation of tubercular granulomas. An in vitro study using MT-infected human leukocytes showed that 1,25(OH)_2_D significantly attenuated MT-induced increases in expression of MMP-7 and MMP-10, and suppressed secretion of MMP-7 by MT-infected PBMC, whilst MMP-9 gene expression, secretion and activity were significantly inhibited by 1,25(OH)_2_D_3_ irrespective of infection [28,29]. It appears that vitamin D and its hydroxylated derivatives also promote the stabilization of the endothelium and of the barrier function in the presence of inflammatory mediators [30], as summarized in Table 2. Moreover, it has been observed that vitamin D, through LL-37, induces a modulation of the expression of pro-inflammatory and anti-inflammatory cytokines (reduction of pro-inflammatory cytokines TNFα and IL-17 and increase of anti-inflammatory ones IL-10 and TGF-β), without reducing the antimycobacterial activity [31]. This is essential to reduce the inflammatory state and tissue damage that characterize the pathophysiology of TB.

## 5. Polymorphisms in VDR

The vitamin D receptor (VDR), after binding the activated vitamin D, forms a heterodimer with the retinoic acid receptor (RXR); then it is internalized in the cell and reaches the nucleus, where it interacts with the transcription factors and binds the gene portion of DNA responsive to vitamin D, called VDRE, thus influencing the expression or repression of genes, which include numerous metabolic and immune pathways and response regulation against cancer.

VDR is a 427 amino-acid protein, encoded by the VDR gene, found on chromosome 12q13.11 in humans. Several polymorphisms have been identified in the 3′ region of the human VDR gene, thanks to the use of restriction endonucleases, the main ones being *TaqI*, *BsmI*, *FokI* and *ApaI*. The alleles of each gene are written with the capital letter (T, B, F) for the absence of the restriction site, while the lowercase letter (t, b, f) is used to indicate its presence. Therefore, homozygous dominant (TT, BB, FF), heterozygous (Tt, Bb, Ff) or homozygous recessive (tt, bb, ff) forms may be present. For the *FokI* polymorphism, for example, the FF form leads to an increase in transcription, *BsmI* is associated with transcript stability and *TaqI* is responsible for a higher cell turnover.

These polymorphisms were the subject of numerous studies regarding TB because of the possible influence of gene variations on the individual response to MT. There are numerous studies in different populations worldwide that show a variation in the response to both disease and therapy for TB. For example, the *FokI*-ff polymorphism has been associated with the extended form of pulmonary TB, in the study conducted on Gujarati Indians living in the UK [21], and with a higher susceptibility for TB among Chinese patients [32]. In another study on patients from Gambia, it was seen how the *TaqI*-tt polymorphism was implicated in the reduced possibility of developing active TB [33], but the same polymorphism on the Gujarati Indians only showed that the presence of non-tt genes gave greater susceptibility, if the blood concentration of active vitamin D was not sufficient (less than 10 nmol/L) [34]. This suggests that the polymorphism does not affect alone on the response to MT, but is a gene-environment vitamin D interaction.

The *FokI* and *TaqI* polymorphisms were also studied in a Peruvian community, where the association with the speed of healing is reported; it was highlighted that patients with Tt or FF polymorphism rather than TT or non-FF achieved more quickly the negativization of sputum culture [35]. Unfortunately, the data cannot be generalized due to the small size of sample and the unclear significance of these gene variants [36].

Furthermore, as already mentioned above, monocytes infected with MT produce enzymes belonging to the metalloproteases of the matrix (MMP-9), which act by degrading the surrounding extracellular matrix; it allows MT to spread more easily to neighboring cells. The study conducted by Timms PM et al. showed a correlation between the T allele of *TaqI* and the reduced production of an antiproteinase (TIMPs) that inhibits MMP-9 [37]. We can therefore say that calcitriol modulates the production of MMP [38]. For this reason, it could be correlated with the evolution and clinical severity of TB.

The results obtained are not only attributable to the interaction between the VDR phenotype and MT, but to a set of factors that can also be environmental, social and behavioral, such as therapy compliance or the socio-economic context in which the patient pool is inserted; for example, it has been seen that a higher level of education was linked with more adherence to treatment and therefore better outcome [35,39].

Some meta-analysis carried out on these different studies concerning the relationship between VDR polymorphisms and TB showed an evident positive association between the ff variant of the *FokI* gene and TB, especially in HIV-negative TB patients and the Asian group [40,41].

On the other hand, a significant inverse association was found with the *BsmI*-bb genotype and a correlation of scarce significance with the polymorphisms of *TaqI* and *ApaI* [40]. Table 3 summarizes the evidence found in the various studies on the correlation between the main polymorphisms of the VDR gene and TB.

## 6. The Role of Vitamin D in Prevention of Tuberculosis

Given the properties and mechanisms of action of vitamin D currently known, several studies have questioned the link between vitamin D levels and the risk of progression to active TB in patients exposed to MT. First, a lower concentration of 25(OH)D has been demonstrated in patients with active TB in comparison with healthy patients [42,43,44]. However, it is not clear whether the vitamin D deficiency is due to infection or whether the progression of the infection is favored by the vitamin D deficiency. Moreover, there is evidence of the ability of vitamin D to inhibit the replication of MT in vitro [26,45]. For example, a study has shown that a concentration of 4 µg/mL of cholecalciferol is sufficient to slow down the proliferation of the bacillus inside cultured human macrophages. This value is significantly higher than the normal circulating levels of 1,25(OH)_2_D; however, as already mentioned, the infected macrophages are capable of autonomously producing this active form of vitamin D and could be therefore capable of reaching such concentrations [45]. On the other hand, the studies carried out in vivo are discordant and have not led to unequivocal conclusions about the efficacy of vitamin D supplementation in preventing the development of the disease.

In a randomized controlled clinical trial conducted on 8851 children in Mongolia with latent tuberculosis infection (LTBI) determined by QuantiFERON(^®^), et al. found no significant difference in the reduction of the risk of tuberculosis infection and tuberculosis disease among children who were given a vitamin D supplementation (14,000 UI of vitamin D for week) and those treated with placebo [46]. However, a meta-analysis, conducted on studies focused on various aspects of the relations between vitamin D and TB, found that a low level of 25(OH)D is associated with an increased risk of developing active TB. On the other hand, the same study showed a trend of higher levels of 1,25(OH)_2_D (the bioactive form) in subjects with active TB, supporting the theory that the 25(OH)D deficiency is due to an increase in its conversion into the bioactive form in response to infection [43]. Another meta-analysis of prospective studies carried out by Aibana O, Huang C-C et al. [47] confirmed the results of the previous one, showing in fact, a positive dose-dependent correlation between pre-existent low levels of 25(OH)D in the bloodstream and an increased risk of developing active disease in high risk groups (LTBI subject/household contacts of active TB patients). This risk was higher among HIV-positive patients with severe vitamin D deficiency. Nevertheless, both meta-analyses have important limitations, including the variety that exists in the definition of vitamin D deficiency among the studies on which they are based. So, there is a need for further studies and clinical trials to evaluate the effectiveness of the vitamin D supplement in the prevention of active TB.

## 7. The Role of Vitamin D in Treatment of Tuberculosis

Given the high spread of TB worldwide, the long duration of therapy, the scarce availability of anti-tuberculosis drugs and the increasing presence of infections caused by resistant to first-line drugs MT, it is necessary to search for new medications or supplements that could reduce the time of administration of therapy and enhance the effect of already existing drugs. Although several studies have confirmed in vitro the important role of active vitamin D against MT [47,48], randomized controlled trials (RCTs) did not confirm what was expected, and the results of the in vivo studies are conflicting, so the debate is still open. Jing Zhang et al. conducted an in vivo study on mice in which the therapeutic synergy between vitamin D and pyrazinamide (PZA) was analyzed. This study demonstrates how using calcitriol and PZA concurrently results in an interruption of bacterial growth and a faster resolution of MT-related lung lesions. Furthermore, it has been shown that the administration of vitamin D during therapy with PZA results in an increase of the release of anti-inflammatory cytokines and antimicrobial molecules, respectively IL-4 and LL-37. The latter would otherwise be reduced in patients taking PZA without vitamin D supplementation [49]. On the other hand, in some RCTs that have evaluated the infusion of high doses of vitamin D, beneficial effects were not observed, and the patients did not reach an earlier microscopic negativization compared to the patients who were given a placebo [50,51]. Of note, despite multiple studies of vitamin D supplementation in different doses, statistically significant benefits on sputum conversion have not been demonstrated [52]. It must be said, however, that few studies have been conducted on this topic, with numerically small samples and data inaccuracies. For these reasons, further clinical trials to evaluate the effective role of the vitamin D supplement in the treatment of TB are needed. One of the hypotheses to explain this difference between the in vivo and in vitro results is the possible interference between the drugs used for the treatment of TB and the metabolism of VD. In this regard, Chesdachai et al. have analyzed this relation through an in vitro study [53]. In this study, human monocytes were cultured with calcitriol and anti-tuberculosis drugs at different concentrations, and the activity of the hCAP18/cathelicidin system was analyzed. The study showed that the culture with lower concentration of INH led to a strong induction of hCAP18/cathelicidin system; Rifampicin at the same concentration resulted in a repression of its expression. The other cultures, at higher concentrations of isoniazid or rifampicin and all cultures with ethambutol or pyrazinamide alone, did not lead to any change in cathelicidin production. The culture with the four drugs, added together at maximum concentration (10 mcg/mL), showed strong inhibition of hCAP18/cathelicidin in presence of 1,25(OH)₂D₃. They have demonstrated that the combination of the four drugs used in the first-line treatment of TB (PZA, INH, ETM, RIF) can inhibit the increase in hCAP18/LL-37 expression induced by the bioactive form of vitamin D in cultured human macrophages [53]. This result obtained in vitro, may explain why in subjects infected with drug-sensitive MT, who are administered full doses of the four anti-tuberculosis drugs, do not obtain rapid improvements with the administration of calcitriol in oral form; in fact, by analyzing both the expression of hCAP18 and the presence of LL-37 in the bloodstream, there are no increases after 8 weeks of administration, compared to the control subjects [51]. Several observational studies have shown the persistence of low levels of vitamin D in patients with active TB [54,55,56], and some have shown a statistically significant relationship with the intake of anti-tuberculosis drugs [54]. The level of vitamin D in the bloodstream depends above all on the hepatic metabolism and its level of impairment; an open question is to determine whether the conventional therapeutic regimen can influence the blood concentrations of the vitamin, because, as it is known, anti-TB antibiotics are characterized by a non-negligible liver toxicity. Some studies underline how the use of INH can change the levels of 25-hydroxylase and 1-hydroxylase, and consequently also of the active VD. This is because INH inhibits or induces the cytochrome P450 system which regulates enzymatic activation [55]. Another evidence concerns RIF, which appears to influence the metabolism, enhancing CYP3A4 (but not CYP24A1), which acts as a 24-hydroxylase for 25(OH)D, reducing the production of the active form of vitamin D [56]. Regarding PZA and ETB, there are no studies that demonstrate changes in vitamin D.

## 8. Synergism with Vitamin A in Prevention and Treatment of TB

A synergistic activity of vitamin D and vitamin A (VA) against MT has been hypothesized but the data regarding its role in treatment are discordant [57]. VA and vitamin D bind to their intracellular receptors, retinoic acid receptor (RAR) and VDR, respectively, and subsequently both bind to the retinoid X receptor (RXR) and mediate changes in gene expression within the cell. A study demonstrated in vitro the synergistic effect of these two vitamins in inhibiting the entry of MT and its survival in macrophages, through the downregulation of the expression of the TACO gene [58]. Some in vivo studies have not found a clear relationship between these two vitamins in the impact on the treatment of TB; however, they seem to suggest an influence of other micronutrients on the effects of vitamin D [47]. For this reason, further studies are needed.

## 9. Side Effects of Vitamin D Supplementation

Potential effects of vitamin D supplementation include hypercalcemia, hypercalciuria and potentially nephrocalcinosis, especially in patients with renal failure. However, a meta-analysis conducted on RCTs investigated the cumulative relative risk of any type of adverse event, as well as kidney stones, hypercalcemia and hypercalciuria following administration of at least 2800 IU/day of vitamin D₂ or D₃ for at least 1 year, concluding that high doses of vitamin D administered for a long time (at least one year) do not significantly increase the risk of adverse events, although there is a trend toward increased calcium and possibly hypercalciuria [59].

## 10. Future Perspectives

Tuberculosis remains a major public health problem worldwide. The increase in cases worldwide and the spread of multidrug-resistant forms make it necessary to search for new tools that help fight the disease. Vitamin D is now known for its multiple properties in human health; however, while its role in mineral homeostasis is well investigated and shared guidelines are available, there are many unanswered questions about its other functions, so the debate is still open. In particular, the in vitro antimycobacterial properties of vitamin D are known, and several mechanisms by which vitamin D exerts this action have been recognized, but the results of in vivo studies regarding its role in the prophylaxis and treatment of TB are inconclusive and discordant. In fact, it is necessary to consider several factors that can influence the variability of the results, including polymorphisms of the genes that code for VDR, interaction with anti-tuberculosis drugs, environmental and social factors, other vitamins and micronutrients deficiencies, the lack of a common definition of vitamin D deficiency and the variability of the methods of its administration among the various studies.

In conclusion, further clinical trials are needed to study its efficacy in vivo, eliminating confounding factors, and to eventually determine which dosages and methods of administration are most suitable in the treatment and prophylaxis of TB. The possible in vivo benefit of a joint administration with other vitamins, such as vitamin A, which share a known in vitro antimycobacterial action with vitamin D [57], should be studied. However, based on the evidence of its efficacy in vitro, considering the low incidence of side effects even at high dosages and its low cost, it would be advisable to assess vitamin D level both in patients with active TB and high-risk groups and administer it, at least to reach sufficient levels.

## Figures and Tables

**Figure 1 ijms-23-03860-f001:**
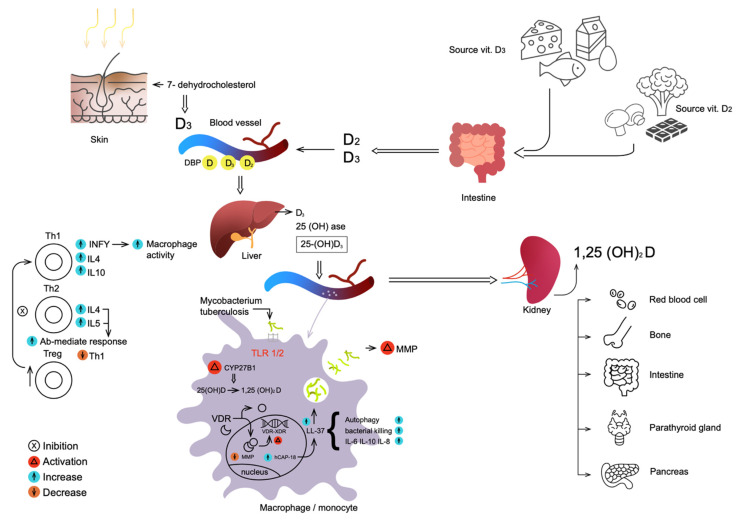
Metabolism and action of Vitamin D in MT infection. Vitamin D is synthesized in the skin following stimulation by UV rays or introduced with food. In the blood, vitamin D circulates free or bound to VDBP, reaches the liver, where it is converted to 25-(OH)-vit. D by the enzyme hydroxylase. Reintroduced into the bloodstream, it reaches the kidney, where it is further hydroxylated; this active form has multiple activities on different organs (parathyroid, bones, pancreas, intestine). Part of the 25-(OH)-vit. D enters the macrophages, where by hydroxylation it is activated, binds the VDR, passes into the nucleus and increases the transcription of genes for hCAP18. This, cleaved, produces LL-37, which has anti-MT activity, increases autophagy and macrophage killing. In addition, 25(OH)D modulates the innate and adaptive immune system and increases in cytokines (IL-4, IL-5), and INF-γ increases in the antibody-mediated response.

**Table 1 ijms-23-03860-t001:** 25(OH)D thresholds and respective serum values.

Serum Value of 25(OH)D (ng/mL)	
30–100	Sufficiency
20–30	Insufficiency
<20	Deficiency

**Table 2 ijms-23-03860-t002:** Summary of antimicrobial and anti-inflammatory actions of Vitamin D.

Pro-Inflammatory/Antimicrobial Actions of Vit. D	Anti-Inflammatory Actions of Vit. D
Induces destruction of the bacterial cell by activating the cathelicidin/LL-37 system in infected macrophages	Induces the expansion of T-reg lymphocytes, which in turn limit the activity of Th1
Induces autophagy in infected macrophages	Attenuates *M. tuberculosis*-induced expression of MMP
Inhibits the growth of MT in infected macrophages through the production of nitrogen and oxygen reactants	Stabilization of the endothelium and of the barrier function in the presence of inflammatory mediators
Stimulates production of INF-γ by Th1 lymphocytes, which enhances antibacterial activity of macrophages	Reduction of pro-inflammatory cytokines and increase of anti-inflammatory ones, without reducing the antimycobacterial activity
Enhances the antibody-mediated response by Th2 lymphocytes via the secretion of IL4 and IL5	

**Table 3 ijms-23-03860-t003:** Correlation between polymorphism in VDR gene and possible effects on TB.

Polymorphism in VDR Gene	Allele’s Possible Combination	Effects
*TaqI*	TT	Higher cell turnover, reduction in TIMPs
	Tt	Increase in culture conversion, quick negativization of sputum
	tt	Reduction of active TB
*Bsml*	BB	Transcript stability
	Bb	-
	bb	Significant association with active TB
*Fokl*	FF	Increase in transcription, quick negativization of sputum
	Ff	-
	ff	Extended form of pulmonary TB

## Data Availability

Not applicable.

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
