# Peer review of "Impact of Vitamin D in Prophylaxis and Treatment in Tuberculosis Patients"

_ijms, 2022, doi:10.3390/ijms23073860_

Round 1
Reviewer 1 Report
Review of Papagni et al, Impact of Vitamin D in prophylaxis and treatment in tuberculosis patients
International Journal of Molecular Sciences
In summary
This review sets out to elucidate the antimicrobial properties of Vitamin D, particularly in the context of tuberculosis. The authors go through some fascinating evidence which is highly interesting, and very worthwhile of publication. However, the paper has some methodological shortcomings which I would encourage the authors to address, to both increase the transparency of how literature was selected, and to ensure that the paper is easier to read.
The authors do not state which type of review, but it appears to be a narrative type review. The authors do not describe their methods or literature search strings. It would be appropriate to follow the protocol for systematic reviews eg. the PRISMA guidelines.
Recommendation: Major revision
Major concerns
- The paper completely lacs a section describing the methods, please see above comment. I suggest to describe in detail how literature search was conducted and, if possible, apply the systematic review approach as given by the PRISMA guidelines.
- The paper also lacks a discussion and conclusions section. It seems to be structured more in the way of a textbook chapter. I would suggest to restructure to a format more commonly applied in scientific articles.
Minor concerns
- Please correct to present tense throughout where appropriate (eg line 73, “Figure shows”, not “Figure showed”)
- I am not sure VD or vit.D are commonly accepted abbreviations for Vitamin D. The nomenclature usually states 25(OH)D. Please check for correct nomenclature and correct throughout.
- Tuberculosis is not with capital T, please correct where appropriate
Author Response
To International Journal of Molecular Sciences editor,
We have appreciated the positive feedback to our manuscript “Impact of Vitamin D in prophylaxis and treatment in tuberculosis patients”. We have considered all the useful suggestions made by the referees and we have implemented the text. We have also satisfied the technical requirements according to the journal guidelines. Modifications have been highlighted using the "track changes" feature. Also, a native English speaker has been engaged to improve the fluency and the readability of the manuscript.
We believe that the revision proposed by the referees, and further implemented in the text, contributed to improve the manuscript. Thus, we kindly ask you to re-consider the manuscript for publication.
Please find a point-by-point response to the referees’ comments below.
Best regards,
Dr. Francesco Di Gennaro
Reviewer 1:
This review sets out to elucidate the antimicrobial properties of Vitamin D, particularly in the context of tuberculosis. The authors go through some fascinating evidence which is highly interesting, and very worthwhile of publication. However, the paper has some methodological shortcomings which I would encourage the authors to address, to both increase the transparency of how literature was selected, and to ensure that the paper is easier to read.
Response: We thank you very much for the encouraging feedback on our manuscript. We followed your suggestions and believe that now the paper is more usable for the scientific community.
The authors do not state which type of review, but it appears to be a narrative type review. The authors do not describe their methods or literature search strings. It would be appropriate to follow the protocol for systematic reviews eg. the PRISMA guidelines.
Response: Thanks for your suggestions. The paper is a literature review and the methods are the following: We conducted a search on PubMed, Scopus, Google Scholar, EMBASE, Cochrane Library and WHO websites starting from March 1950 to February 2022, in order to identify articles discussing the role of Vitamins D and TB. We included all articles addressing epidemiology, physiopathology, clinical features, screening and diagnosis, treatment and management.
But, as suggested by editor, we removed methods section during our submission
Major concerns
- The paper completely lacs a section describing the methods, please see above comment. I suggest to describe in detail how literature search was conducted and, if possible, apply the systematic review approach as given by the PRISMA guidelines.
- The paper also lacks a discussion and conclusions section. It seems to be structured more in the way of a textbook chapter. I would suggest to restructure to a format more commonly applied in scientific articles.
Response: Many thanks for your suggestions. The paper is a literature review and the methods are the following: We conducted a search on PubMed, Scopus, Google Scholar, EMBASE, Cochrane Library and WHO websites starting from March 1950 to February 2022, in order to identify articles discussing the role of Vitamins D and TB. We included all articles addressing epidemiology, physiopathology, clinical features, screening and diagnosis, treatment and management.
But, as suggested by editor, we removed methods section during our submission. Furthermore, following your suggestion we add a discussion section.
Minor concerns
- Please correct to present tense throughout where appropriate (eg line 73, “Figure shows”, not “Figure showed”)
- I am not sure VD or vit.D are commonly accepted abbreviations for Vitamin D. The nomenclature usually states 25(OH)D. Please check for correct nomenclature and correct throughout.
- Tuberculosis is not with capital T, please correct where appropriate
Response: Many thanks for your suggestions. We correct the manuscript following your suggestions. About Vitamin D we, according with your comment we uniformed the nomenclature of Vitamin D in 1,25(OH)â‚‚D or Vitamin D.
Reviewer 2 Report
Authors discuss an interesting scientific issue, narrating the relevant literature in a structured manner. Yet, their conclusions may not be validated as qualitative and quantitave evaluation of the literature is missing.
PRISMA protocol should be followed and the results should be presented in a constructive way. If meta-analysis is not possible, authors should explain the reason.
Author Response
To International Journal of Molecular Sciences editor,
We have appreciated the positive feedback to our manuscript “Impact of Vitamin D in prophylaxis and treatment in tuberculosis patients”. We have considered all the useful suggestions made by the referees and we have implemented the text. We have also satisfied the technical requirements according to the journal guidelines. Modifications have been highlighted using the "track changes" feature. Also, a native English speaker has been engaged to improve the fluency and the readability of the manuscript.
We believe that the revision proposed by the referees, and further implemented in the text, contributed to improve the manuscript. Thus, we kindly ask you to re-consider the manuscript for publication.
Please find a point-by-point response to the referees’ comments below.
Best regards,
Dr. Francesco Di Gennaro
Reviewer 2:
Authors discuss an interesting scientific issue, narrating the relevant literature in a structured manner. Yet, their conclusions may not be validated as qualitative and quantitave evaluation of the literature is missing.
PRISMA protocol should be followed and the results should be presented in a constructive way. If meta-analysis is not possible, authors should explain the reason.
Response: We thank you very much for the encouraging feedback on our manuscript. We followed your suggestions and believe that now the paper is more usable for the scientific community.
The performed a literature review and the methods are the following: We conducted a search on PubMed, Scopus, Google Scholar, EMBASE, Cochrane Library and WHO websites starting from March 1950 to February 2022, in order to identify articles discussing the role of Vitamins D and TB. We included all articles addressing epidemiology, physiopathology, clinical features, screening and diagnosis, treatment and management. But, as suggested by editor, we removed methods section during our submission. Furthermore, following your suggestion we add a section “Future perspectives” that include our reflections and conclusions on the role of Vitamin D in Tuberculosis.